# Dungeons and Data: A Large-Scale NetHack Dataset

**Eric Hambro**∗
Meta AI

**Roberta Raileanu**
Meta AI

**Danielle Rothermel**
Meta AI

**Vegard Mella**
Meta AI

**Tim Rocktäschel**
University College London†

**Heinrich Küttler**
Inflection AI†

**Naila Murray**
Meta AI

## Abstract

Recent breakthroughs in the development of agents to solve challenging sequential decision making problems such as Go [50], StarCraft [58], or DOTA [3], have relied on both simulated environments and large-scale datasets. However, progress on this research has been hindered by the scarcity of open-sourced datasets and the prohibitive computational cost to work with them. Here we present the NetHack Learning Dataset (`NLD`), a large and highly-scalable dataset of trajectories from the popular game of NetHack, which is both extremely challenging for current methods and very fast to run [23]. `NLD` consists of three parts: 10 billion state transitions from 1.5 million human trajectories collected on the `NAO` public NetHack server from 2009 to 2020; 3 billion state-action-score transitions from 100,000 trajectories collected from the symbolic bot winner of the NetHack Challenge 2021; and, accompanying code for users to record, load and stream any collection of such trajectories in a highly compressed form. We evaluate a wide range of existing algorithms including online and offline RL, as well as learning from demonstrations, showing that significant research advances are needed to fully leverage large-scale datasets for challenging sequential decision making tasks.

## 1 Introduction

Recent progress on deep reinforcement learning (RL) methods has led to significant breakthroughs such as training autonomous agents to play Atari [33], Go [50], StarCraft [58], and Dota [3], or to perform complex robotic tasks [24, 44, 26, 35, 16, 41]. In many of these cases, success relied on having access to large-scale datasets of human demonstrations [58, 3, 41, 26]. Without access to such demonstrations, training RL agents to operate effectively in these environments remains challenging due to the hard exploration problem posed by their vast state and action spaces. In addition, having access to a simulator is key for training agents that can discover new strategies not exhibited by human demonstrations. Therefore, training agents using a combination of offline data and online interaction has proven to be a highly successful approach for solving a variety of challenging sequential decision making tasks. However, this requires access to complex simulators and large-scale offline datasets, which tend to be computationally expensive.

The NetHack Learning Environment (`NLE`) was recently introduced as a testbed for RL, providing an environment which is both extremely challenging [15] and exceptionally fast to simulate [23]. `NLE` is a stochastic, partially observed, and procedurally generated RL environment based on the popular game of NetHack. Due to its long episodes (*i.e.* tens or hundreds of thousands of steps) and large state and action spaces, `NLE` poses a uniquely hard exploration challenge for current RL methods. Thus, one of the most promising research avenues towards progress on NetHack is leveraging human or symbolic-bot demonstrations to bootstrap performance, which also proved successful for StarCraft [58] and Dota [3].

---

∗Correspondence to ehambro@fb.com.
†Work done while at Meta AI.

36th Conference on Neural Information Processing Systems (NeurIPS 2022) Track on Datasets and Benchmarks.

In this paper, we introduce the NetHack Learning Dataset (NLD), an open and accessible dataset for large-scale offline RL and learning from demonstrations. NLD consists of three parts: first, NLD-NAO: a collection of state-only trajectories from 1.5 million human games of NetHack played on nethack.alt.org (NAO) servers between 2009 and 2020; second, NLD-AA: a collection of state-action-score trajectories from 100,000 NLE games played by the symbolic-bot winner of the 2021 NetHack Challenge [15]; third, TtyrecDataset: a highly-scalable tool for efficient training on any NetHack and NLE-generated trajectories and metadata. NLD, in combination with NLE, enables computationally-accessible research in multiple areas including imitation learning, offline RL, learning from sequences of only observations, as well as combining learning from offline data with learning from online interactions. In contrast with other large-scale datasets of demonstrations, NLD is highly efficient in both memory and compute. NLD-NAO can fit on a $30 hard drive, after being compressed (by a factor of 160) from 38TB to 229GB. In addition, NLD-NAO can be processed in under 15 hours, achieving a throughput of 288,000 frames per second with only 10 CPUs. NLD's low memory and computational requirements makes large-scale learning from demonstrations more accessible for academic and independent researchers.

To summarize, the key characteristics of NLD are that: it is a **scalable dataset of demonstrations** (*i.e.* large and cheap) for a highly-complex sequential decision making challenge; it enables **research in multiple areas** such as imitation learning, offline RL, learning from observations of demonstrations, learning from both static data and environment interaction; and it has **many properties of real-world domains** such as partial observability, stochastic dynamics, sparse reward, long trajectories, rich environment, diverse behaviors, and a procedurally generated environment.

In this paper, we make the following core contributions: (i) we introduce NLD-NAO, a large-scale dataset of almost 10 billion state transitions, from 1.5 million NetHack games played by humans; (ii) we also introduce NLD-AA, a large-scale dataset of over 3 billion state-action-score transitions, from 100,000 games played by the symbolic-bot winner of the NeurIPS 2022 NetHack Competition; (iii) we open-source code for users to record, load, and stream any collection of NetHack trajectories in a highly compressed form; and (iv) we show that, while current state-of-the-art methods in offline RL and learning from demonstrations can effectively make use of the dataset, playing NetHack at human-level performance remains an open research challenge.

## 2 Related Work

**Offline RL Benchmarks.** Recently, there has been a growing interest in developing better offline RL methods [25, 39, 11, 60, 61, 21, 20, 6, 2, 49] which aim to learn from datasets of trajectories. With it, a number of offline RL benchmarks have been released [1, 62, 10, 42, 22, 43]. While these benchmarks focus specifically on offline RL, our datasets enable research on multiple areas including imitation learning, learning from observations only (*i.e.* without access to actions or rewards), as well as learning from both offline and online interactions. In order to make progress on difficult RL tasks such as NetHack, we will likely need to learn from both human data and environment interaction, as was the case with other challenging games like StarCraft [58] or Dota [3]. In contrast, the tasks proposed in current offline RL benchmarks are much easier and can be solved by training either only online or only offline [10, 62]. In addition, current offline RL benchmarks test agents on the exact same environment where the data was collected. As emphasized by [57], imitation learning algorithms drastically overfit to their environments, so it is essential to evaluate them on new scenarios in order to develop robust methods. In contrast, NLE has long procedurally generated episodes which require both long-term planning and systematic generalization in order to succeed. This is shown in [30], which investigates transfer learning between policies trained on different NLE-based environments.

For many real-world applications such as autonomous driving or robotic manipulation, learning from human data is essential due to safety concerns and time constraints [40, 10, 8, 5, 27, 52, 55, 17, 38, 18]. However, most offline RL benchmarks contain synthetic trajectories generated by either random exploration, pretrained RL agents, or simple hard-coded behaviors [10]. In contrast, one of our datasets consists entirely of human replays, while the other one is generated by the winner of the NetHack Competition at NeurIPS 2022 which is a complex symbolic bot with built-in knowledge of the game. Human data (like the set contained in NLD-NAO) is significantly more diverse and messy than synthetic data, as humans can vary widely in their expertise, optimize for different objectives (such as fun or discovery), have access to external information (such as the NetHack Wiki [34]),

and even have different observation or action spaces than RL agents. Hence, learning directly from human data is essential for making progress on real-world problems in RL.

**Large-Scale Human Datasets.** A number of large-scale datasets of human replays have been released for StarCraft [59], Dota [3], and MineRL [14]. However, training models on these datasets requires massive computational resources, which makes it unfeasible for academic or independent researchers. In contrast, `NLD` strikes a better balance between scale (*i.e.* a large number of diverse human demonstrations on a complex task) and efficiency (*i.e.* cheap to use and fast to run).

For many real-world applications such as robotic manipulation, we only have access to the demonstrator's observations and not their actions [29, 48, 40, 8, 51]. Research on this setting has been slower [9, 56, 7], in part due to the lack of efficient large-scale datasets. While there are some datasets containing only observations, they are either much smaller than `NLD` [32, 57, 48], too computationally expensive [59, 3], or lack a simulator which prevents learning from online interactions [12, 28, 8].

## 3  Background: The NetHack Learning Environment

The NetHack Learning Environment (`NLE`) is a *gym* environment [4] based on the popular "dungeon-crawler" game, NetHack [19]. Despite the visual simplicity, NetHack is widely considered one of the hardest video games in history since it can take years for humans to win the game [54]. Players need to explore the dungeon, manage their resources, as well as learn about the many entities and their dynamics (often by relying on external knowledge sources like the NetHack Wiki [34]). NetHack has a clearly defined goal, namely descend the dungeon, retrieve an amulet, and ascend back to win the game. At the beginning of each game, players are randomly assigned a given multidimensional character defined by role, race, alignment, and gender (which have varying properties and challenges), so they need to master all characters in order to win consistently. Thus, `NLE` offers a unique set of properties which make it well-positioned to advance research on RL and learning from demonstrations: it is a highly complex environment, containing hundreds of entities with different dynamics; it is procedurally generated, allowing researchers to test generalization; it is partially observed, highly stochastic, and has very long episodes (*i.e.* one or two orders of magnitude longer that Starcraft II [59]).

Following its release, several works have built on `NLE` to leverage its complexity in different ways. MiniHack [45] allows researchers to design their own environments to test particular capabilities of RL agents, by leveraging the full set of entities and dynamics from NetHack. The NetHack Challenge [15] was a competition at NeurIPS 2021, which sought to incentivise a showdown between symbolic and deep RL methods on `NLE`. Symbolic bots decisively outperformed deep RL methods, with the best performing symbolic bots surpassing state-of-the-art deep RL methods by a factor of 5.

## 4  The NetHack Learning Dataset

The NetHack Learning Dataset (`NLD`) contains three components:

1. `NLD-NAO` — a directory of `ttyrec.bz2` files containing almost 10 billion state trajectories and metadata from 1,500,000 human games of NetHack played on `nethack.alt.org`.
2. `NLD-AA` — a directory of `ttyrec3.bz2` files containing over 3 billion state-action-score trajectories and metadata from 100,000 games collected from the winning bot of the NetHack Challenge [15].
3. `TtyrecDataset` — a Python class that can scalably load directories of `ttyrec.bz2` / `ttyrec3.bz2` files and their metadata into `numpy` arrays.

We are also releasing a new version of the NetHack environment, `NLE v0.9.0`, which contains new features and ensures compatibility with `NLD` (see Appendix C).

**File Format.** The `ttyrec`[3] file format stores sequences of terminal instructions (equivalent to observations in RL), along with the times at which to play them. In `NLE v0.9.0`, we adapt this format to also store keypress inputs to the terminal (equivalent to actions in RL), and in-game scores over time (equivalent to rewards in RL), allowing a reconstruction of state-action-score trajectories. This

---

[3] https://nethackwiki.com/wiki/Ttyrec

adapted format is known as `ttyrec3`. The `ttyrec.bz2` and `ttyrec3.bz2` formats, compressed versions of `ttyrec` and `ttyrec3`, are the primary data formats used in NLD. Using `TtyrecDataset`, these compressed files can be written and read on-the-fly, resulting in data compression ratios of more than 138. The files can be decompressed into the state trajectory on a terminal, by using a terminal emulator and querying its screen. For more details see Appendix D.

Throughout the paper, we refer to a player's input at a given time as either state or observation. However, note that NetHack is partially observed, so the player doesn't have access to the full state of the game. We also sometimes use the terms score and reward interchangeably, since the increment in in-game score is a natural choice for the reward used to train RL agents on NetHack. Similarly, a human's keypress corresponds to an agent's action in the game.

**Metadata.** NetHack has an optional built-in feature for the logging of game metadata, used for the maintenance of all-time high-score lists. At the end of a game, 26 fields of data are logged to a common `xlogfile`[4] for posterity. These fields include the character's *name*, *race*, *role*, *score*, *cause of death*, *maximum dungeon depth*, and more. See Appendix E for more details on these fields. With `NLE v0.9.0`, an `xlogfile` is generated for all NLE recordings. These files are used to populate all metadata contained in NLD.

**State-Action-Score Transitions.** As mentioned, `NLD-AA` contains sequences of state-action-score transitions from symbolic-bot plays, while `NLD-NAO` contains sequences of state transitions from human plays. These transitions are efficiently processed using the `TtyrecDataset`. The states consist of: `tty_chars` (the characters at each point on the screen), `tty_colors` (the colors at each point on the screen), `tty_cursor` (the location of the cursor on the screen), `timestamps` (when the state was recorded), and `gameids` (an identifier for the game in question). Additionally, `keypresses` and `score` observations are available for `ttyrec3` files, as in the `NLD-AA` dataset. The states, keypresses, and scores from NLD map directly to an agent's observations, actions, and rewards in NLE. More information about these transitions can be found in Appendix F.

**API.** The `TtyrecDataset` follows the API of an `IterableDataset` defined in PyTorch. This allows for the batched streaming of `ttyrec.bz2` / `ttyrec3.bz2` files directly into fixed NumPy arrays of a chosen shape. Episodes are read sequentially in chunks defined by the unroll length, batched with a different game per batch index. The order of these games can be shuffled, predetermined or even looped to provide a never-ending dataset. This class allows users to load any state-action-score trajectory recorded from `NLE v0.9.0` onwards.

The `TtyrecDataset` wraps a small sqlite3 database where it stores metadata and the paths to files. This design allows for the simple querying of metadata for any game, along with the dynamic subselection of games streamed from the `TtyrecDataset` itself. For example, in Figure 1, we generate sub-datasets from `NLD-NAO` and `NLD-AA`, selecting trajectories where the player has completed the game ('Ascended') or played a 'Human Monk' character, respectively. Appendix G shows how to load only a subset of trajectories, for example where the player has ascended, or a certain role has been used.

**Scalability.** The `TtyrecDataset` is designed to make our large-scale NLD datasets accessible even when the computational resources are limited. To that end, several optimizations are made to improve the memory efficiency and throughput of the data. Most notably, `TtyrecDataset` streams recordings directly from their compressed `ttyrec.bz2` files. This format compresses the 30TB of frame data included in `NLD-NAO` down to 229GB, which can fit on a $30 SSD[5]. This decompression requires on-the-fly unzipping and terminal emulation. The `TtyrecDataset` performs these in GIL-released C/C++. This process is fast and trivially parallelizable with Python Threadpool, resulting in throughputs of over 1.7 GB/s on 80 CPUs. This performance allows the processing of almost 10 billion frames of `NLD-NAO` in 15 hours, with 10 CPUs. See Table 1 for a quantitative description of our two datasets.

## 5 Dataset Analysis

In this section we perform an in-depth analysis of the characteristics of `NLD-AA` and `NLD-NAO`.

---

[4]https://nethackwiki.com/wiki/Xlogfile
[5]https://www.amazon.com/HP-240GB-Internal-Solid-State/dp/B09KFHTYWH

Table 1: `NLD-AA` and `NLD-NAO` in numbers.

|  | NLD-AA | NLD-NAO |
|---|---|---|
| Episodes | 109,545 | 1,511,228 |
| Transitions | 3,481,605,009 | 9,858,127,896 |
| Policies (Players) | 1 | 48,454 |
| Policies Type | symbolic bot | human |
| Transition | (state, action, score) | state |
| Disk Size (Compressed) | 96.7 GB | 229 GB |
| Data Size (Uncompressed) | 13.4 TB | 38.0 TB |
| Compression Ratio | 138 | 166 |
| Mean Epsiode Score | 10,105 | 127,218 |
| Median Episode Score | 5,422 | 836 |
| Median Episode Transitions | 28,181 | 1,724.0 |
| Median Episode Game Turns | 20,414 | 3,766 |
| Epoch Time (10 CPUs) | 4h 49m | 14h 37m |

## 5.1 `NLD-AA`

To our knowledge, `AutoAscend`[6] is currently the best open-sourced artificial agent for playing NetHack `3.6.6`, having achieved first place in the 2021 NeurIPS NetHack Challenge by a considerable margin [15]. `AutoAscend` is a symbolic bot, forgoing any neural network and instead relying on a priority list of hard-coded subroutines. These subroutines are complex, context dependant, and make significant use of NetHack domain knowledge and all NetHack actions. For instances, the bot keeps track of multiple properties for encountered entities and can even solve challenging puzzles such as Sokoban. A full description of `AutoAscend`'s algorithm and behavior can be found in the NetHack Challenge Report [15].

`NLD-AA` was generated by running `AutoAscend` on the `NetHackChallenge-v0` [15] task in NLE `v0.9.0`, utilising its built-in recording feature to generate `ttyrec3.bz2` files. It consists of over 3 billion state-action-score transitions, drawn from 100,000 episodes generated by `AutoAscend`. Of the NLE tasks, `NetHackChallenge-v0` most closely reproduces the full game of NetHack `3.6.6`, and was introduced in NLE `v0.7.0` to grant NetHack Challenge competitors access to the widest possible action space, and force an automated randomisation of the starting character (by race, role, alignment, and gender). By virtue of using `ttyrec3.bz2` files, in-game scores and actions (in the form of keypresses) are stored along with the states, and metadata about the episodes.

**Dataset Skill.** The `AutoAscend` trajectories demonstrate a strong and reliable NetHack player, far exceeding all the deep learning based approaches, but still falling short of an expert human player. NetHack broadly defines a character with less than 2000 score as a 'Beginner'[7]. `AutoAscend` comfortably surpasses the 'Beginner' classification in more than 75% of games for all roles, and in 95% of games for easier roles like Monk (see Figure 1). Given the high variance nature of NetHack games, and the challenge of playing with the more difficult roles, this is an impressive feat.

Compared to the human players in `NLD-NAO`, `AutoAscend`'s policy finds itself just within the top 15% of all players when ranked by mean score. When ranked by median score[8] it comes within the top 7%. However, these metrics are somewhat distorted by the long tail of dilettante players who played only a few games. If instead we define a 'competent' human player as one to have *ever* advanced beyond the Beginner classification, then `AutoAscend` ranks in the top 33% of players by mean score, and top 15% by median.

The competence of `AutoAscend` contrasts both with the poor performance of deep RL bots, and with the exceptional performance required to beat the game. As the winning symbolic bot of the NetHack Challenge, `AutoAscend` beat the winning deep learning bot by almost a factor of 3 in median score, and close to a factor of 5 in mean score. This performance is far outside what deep RL agents can currently achieve, and in some domains `NLD-AA` may be considered an "expert" dataset. However,

---

[6] https://github.com/maciej-sypetkowski/autoascend
[7] https://nethackwiki.com/wiki/Beginner
[8] Median score was the primary metric in the NetHack Challenge

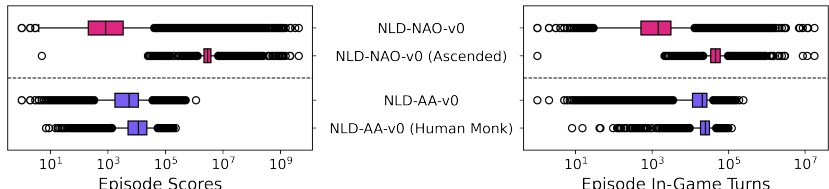

Figure 1: Box plots of episode scores (left) and episode turns (right) for `NLD-NAO` and `NLD-AA` as well as for their corresponding subsets that contain only winning games ('Ascended') or games played with a 'Human Monk' character, respectively. These plots show the distribution in ability in `NLD-NAO` is heavy-tailed, along with the typical scores and game length needed to win, well above the median. By comparison, `NLD-AA` games are competent but not expert, even for easy characters like 'Human Monk'.

when compared to the distribution of winning games (*i.e.* those that have 'ascended'), both mean and median score are orders of magnitude short.

**Dataset Coverage.** Thanks to its scale, `NLD-AA` provides solid coverage of the early-to-mid game of NetHack. `NLD-AA` also provides good coverage of all possible actions and roles, with episodes that last for tens of thousands of transitions. Nevertheless, the dataset contains many complex and diverse behaviors. For the full breakdown of the symbolic bot's behaviors found in `NLD-AA` see Appendix E.

**Datasest Noise.** Since all trajectories were generated using `NLE`, these trajectories have no terminal rendering noise. `NLE` uses a VT100 emulator to record `ttyrec3.bz2` files, and `NLD` uses the same emulator to load them.

### 5.2 `NLD-NAO`

The NetHack community has a long history of using public servers, wherein players can play together, stream and rewatch games of NetHack.[9] The `nethack.alt.org` server, known as `NAO`[10], is one of the longest running of such servers, having started in 2001, and hosts recordings dating back to 2009. The server attempts to record all games in `ttyrec.bz2` format, and hosts them publicly on an S3 bucket[11]. Unlike `NLE`, `NAO` offers players the option of saving a game and resuming play later, allowing for games that span multiple sessions, or last potentially days, weeks, months or years.

`NLD-NAO` was collected by downloading and filtering a dump of `ttyrec.bz2` recordings and metadata from NAO. It consists of almost 10 billion state transitions, drawn from approximately 1.5 million games played by just under 50,000 humans of mixed ability. These games were played between 2009 and 2020, and span NetHack versions `v3.4.0` to `3.6.6`. Since `NAO` provides no link between episode metadata and available `ttyrec` recordings, we developed an algorithm to assign and correctly order `ttyrec` recordings to their corresponding episode metadata, and to filter out trivial or empty games. The full procedure is outlined in Appendix H.

**Dataset Skill.** The `NAO` trajectories were generated from humans of very different skill levels, ranging from novice to expert, and cover a vast distribution of outcomes. The distributions of episode scores, episode turns, and average player scores are highly Zipfian, with long correlated tails spanning several orders of magnitude. The correlation of these distributions greatly improves the quality of the dataset. While the median episode score does not progress beyond Beginner status, such short episodes contribute relatively few transitions to the overall dataset. The bulk of the transitions comes from medium-to-high scoring games that last a long time. The `NLD-NAO` dataset therefore combines two desirable properties: it contains a large number of diverse behaviors with varying levels of performance, as well as many good trajectories from expert demonstrators.

**Dataset Coverage.** Thanks to its size and expert policies, `NLD-NAO` provides an unparalleled coverage of the game of NetHack. The dataset contains many examples of advanced game sections not yet achievable by symbolic or RL bots, such as descending the dungeon and retrieving the amulet. Most importantly, it includes 22,000 examples of winning the game ("Ascension"), and includes several

---

[9]`https://nethackwiki.com/wiki/Public_server`
[10]`https://nethackwiki.com/wiki/Nethack.alt.org`
[11]`s3://altorg/ttyrec/`

examples of doing so with special conducts such as "Nudist" (not wearing any armour). The full distribution of such achievements and conducts can be found in Appendix E.

**Dataset Noise.** Unlike `NLE`, `NAO` does not enforce any fixed terminal type on the player, nor any fixed screen dimensions or configuration. Variations in all these can lead to rendering noise that may include odd symbols or cropped layouts. These types of artefacts are common in real-world applications of learning from demonstrations, which makes `NLD` a suitable dataset for advancing research in this area. More details about these artefacts, as well as other types of noise (*e.g.* matching of recordings to players), see Appendix H.

## 6 Experiments

### 6.1 Methods

We demonstrate the utility and challenge of `NLD` by evaluating algorithms from a range of fields including online RL, offline RL, imitation learning, and learning from demonstrations.

All methods are implemented using the open-source asynchronous RL library `moolib` [31], allowing for continual evaluation of agents in order to generate training curves (even in the offline RL setting). In addition, all our methods use the same core model architecture, which is based on a popular open-source Sample Factory baseline[12], adapted to only use `tty_*` observations but still remain fast to run. Unless specified otherwise, all experiments were evaluated with 1024 episodes from NLE's *NetHackChallenge-v0* task after training for 1.5B frames. The mean and standard deviation of the episode return was computed over 10 training seeds. Full details about the algorithms and their hyperparameters can be found in Appendix I. All our implementations will be open-sourced to enable fast research progress.

**Online RL.** For online RL baselines that do not make use of the `NLD` datasets, we use Asynchronous Proximal Policy Optimization (APPO) [47, 37] and Deep Q-Networks (DQN-Online). The former is a strong on-policy policy-gradient-based method for `NLE`, and the latter is a popular off-policy value-based method in the wider literature.

**Offline RL.** Since `NLD-AA` contains state-action-reward transitions, we can train a number of offline RL baselines on it. More specifically, we use a Deep Q-Network baseline trained on the dataset (DQN-Offline), as well as two state-of-the-art offline RL methods, Conservative Q-Learning (CQL) [21] and Implicit Q-Learning (IQL) [20]. Note that these methods cannot be trained on `NLD-NAO` since it doesn't contain actions or rewards.

**Learning from Demonstrations.** We also evaluate a number of baselines that do not make use of rewards, but instead use state-action or state-only transitions. For `NLD-AA`, we use Behavioural Cloning (BC), a popular imitation learning method, which trains a policy to match the corresponding actions in the dataset using a supervised learning objective. For `NLD-NAO`, we use Behavioral Cloning from Observations (BCO) [56] which is a popular baseline for learning from state-only demonstrations. This method performs BC on a learned inverse dynamics model which was trained to predict the action taken given two consecutive states. We then investigate the use of BC and BCO in an online setting by augmenting our APPO baseline in two ways. First, we add the BC or BCO loss between the agent's policy and the expert's action for observations in the dataset resulting in APPO + BC and APPO + BCO respectively. Second, we add a KL-divergence loss between the agent's policy and a pretrained BC(O) model resulting in Kickstarting BC(O), similar to [46]. These methods make use of state and action data but no rewards.

For `NLD-AA`, we investigate the performance of the above algorithms across two partitions: one with a single character ('Human Monk'), and one with all the characters (*i.e.* the full dataset). Each of these is evaluated on their corresponding environments. Full details about our experimental setup are available in Appendix I.

### 6.2 Results and Discussion

As Table 2 and Figure 2 show, `NLD-AA` poses a difficult challenge to modern offline RL algorithms. While CQL is able to use the offline trajectories to outperform both a random policy, IQL, as well as

---

[12]https://github.com/Miffyli/nle-sample-factory-baseline

Table 2: Average episode return across 10 seeds for a number of popular baselines using NLD and / or NLE. First, we evaluate a range of offline RL methods such as DQN-Offline, CQL, IQL and BC trained on NLD-AA and NLD-AA-Monk, as well as BCO trained on NLD-NAO. Second, we evaluate methods that combine offline and online learning such as Kickstarting + BC and APPO + BC trained on NLD-AA and NLD-AA-Monk, as well as Kickstarting + BCO and APPO + BCO trained on NLD-NAO. Finally, we evaluate a few methods that only have access to online interactions from NLE (no external datasets) such as a Random Policy, DQN-Online, and APPO. All evaluations are conducted with NLE's *NetHackChallenge-v0* task, using Random (@) characters for NLD-AA and NLD-NAO, and a Human Monk character for NLD-AA-Monk. We compare all these to the average episode return in the corresponding datasets. Methods that use both offline and online data outperform all others by a wide margin, but are still significantly worse than humans (by almost two orders of magnitude) and even the symbolic bot (by more than a factor of three). The top online algorithm, APPO, outperforms the top offline approach, BC. These results indicate that NLD poses a substantial challenge to state-of-the-art methods.

| | Character | NLD-AA
Random | NLD-AA-Monk
Human Monk | NLD-NAO
Random |
|---|---|---|---|---|
| offline only | DQN-Offline | 0.0 ± 0.1 | 0.0 ± 0.1 | - |
| | CQL | 704 ± 112 | 732 ± 219 | - |
| | IQL | 341 ± 40 | 533 ± 178 | - |
| | **BC(O)** | **1220 ± 261** | **1728 ± 615** | **12.5 ± 2.0** |
| offline + online | Kickstarting + BC(O) | 2250 ± 81 | 3907 ± 330 | 502 ± 62 |
| | **APPO + BC(O)** | **2767 ± 284** | **4779 ± 1116** | **1972 ± 180** |
| online only | Random Policy | 0.2 ± 0.2 | 0.4 ± 0.4 | (see col 1) |
| | DQN-Online | 159 ± 68 | 247 ± 152 | (see col 1) |
| | **APPO** | **1799 ± 156** | **4006 ± 323** | **(see col 1)** |
| | **Dataset Average** | **10105** | **17274** | **127356** |

online and offline variants of DQN, it falls short of capturing the average performance of trajectories in the dataset by over an order of magnitude. BC performs better due to the higher quality of the supervised learning signal, which is typically denser and less noisy than the RL one, but it too falls well short of the data generating policy. Prior work has shown that offline RL methods struggle to perform well in stochastic environments [36], and to generalize outside the distribution of state-action pairs in the dataset [13]. These challenges are particularly intensified with NLD, which is highly stochastic, partially observed, procedurally generated, and has a very large state and action space.

Methods that combine offline and online training such as Kickstarting + BC(O) or APPO + BC(O) outperform offline-only methods and outperform or match online-only methods, while also speeding up training. In fact, APPO + BC sets a new state-of-the-art for purely Deep RL on NLE, and would have won first place on the Neural Track at the NetHack Challenge 2021, scoring a median of 2110 and beating the hybrid symbolic-deep RL winner by almost 400 points [15]. Despite being our best method, APPO + BC is still significantly worse than the average performance in the dataset, further emphasizing the difficulty of NLD.

Finally, our results demonstrate the utility of NLD-NAO while highlighting the challenge of learning from sequences of only observations. Our inverse model is trained with trajectories from a random policy, and therefore experiences an extremely small fraction of the entire state-action space. Even so, BCO on NLD-NAO labeled with these actions is able to yield performance that comfortably exceeds that of a random policy. While APPO + BCO performs similarly to APPO, training appears to be much more stable. Thus, using more sophisticated policies for training the inverse model (*e.g.* a mix between a pretrained RL policy and a random policy or an exploration method that maximizes the diversity of visited states and actions) could further improve performance.

## 7   Conclusion

In this paper, we introduce NLD, a large-scale dataset of demonstrations from NetHack, a complex, stochastic, partially observed, procedurally generated video game. NLD enables research in multiple

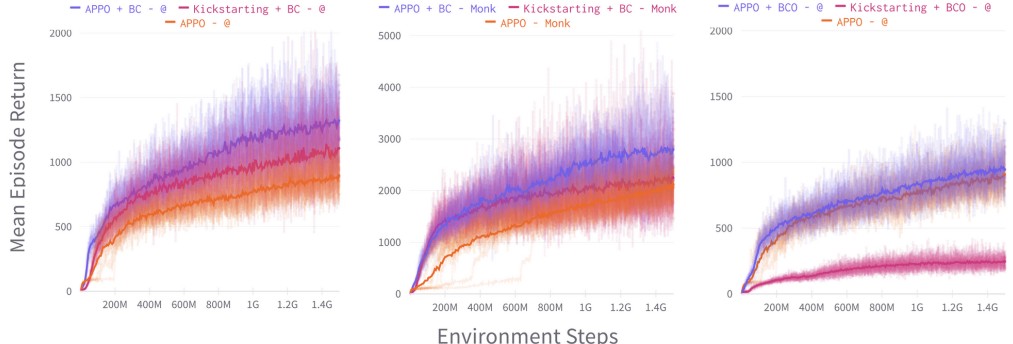

Figure 2: Mean episode return (with samples plotted) over 10 seeds during training for APPO, APPO + BC(O), and Kickstarting + BC(O) using `NLD-AA` (left), `NLD-AA-Monk` (center), and `NLD-NAO` (right). `NLD-AA` and `NLD-NAO` both are evaluated with `Random` character (`@`), while `NLD-AA-Monk` is evaluated with only the `Human Monk` character. Some sample curves associated with the pure online APPO methods exhibit a behaviour wherein performance appears to plateau, before bursting rapidly to a new performance level. This is a common pattern in `NLE` and corresponds to an agent learning a new behaviour that greatly unlocks yet more of the game, such as praying when starving or kicking down locked doors. Since `NLE` is highly procedurally generated with a vast action and state space, it is hard to predict when an agent will encounter sufficient opportunities to "take off" in this way. Such "taking off" phenomena are less visible when learning from `NLD`, as useful behaviours are present in the data throughout.

areas such as imitation learning, offline RL, learning from sequences of observations, and learning from both offline data and online interactions. `NLD` allows researchers to learn from very large datasets of demonstrations without requiring extensive memory or compute resources. We train a number of state-of-the-art methods in online RL (both on-policy and off-policy), offline RL, imitation learning, as well as learning from observation-only sequences. Our results indicate that significant research advances are needed to fully leverage large-scale datasets in order to solve challenging sequential decision making problems.

In many real-world applications such as robotics, it is common to have access to datasets of observations, without corresponding actions or rewards. Thus, one important direction for future research is to learn from trajectories containing only observations. `NLD` enables this type of research in a safe, cheap, fast, and reproducible environment, while still exhibiting many properties of real-world domains such as partial observability, high stochasticity, and long episodes. Due to NetHack being procedurally generated, in order to consistently win the game, an agent needs to learn a robust policy that generalizes to new scenarios. Our strongest baseline learns from both static data and environment interaction, so a promising avenue for future work is to develop more effective methods that combine offline and online learning. Existing algorithms for offline RL and learning from demonstrations struggle to match the average performance of trajectories in our datasets, suggesting that `NLD` is a good benchmark to probe the limits of these methods and inspire the development of approaches that are effective in more challenging and realistic domains.

## 8 Acknowledgements

We would like to thank Drew Streib and the NAO maintainers for their work in the NetHack community and their help throughout this project. We would also like to thank Michał Sypetkowski and Maciej Sypetkowski for their work on the AutoAscend bot.

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

## 10 Datasets Checklist

Include extra information in the appendix. This section will often be part of the supplemental material. Please see the call on the NeurIPS website for links to additional guides on dataset publication.

1. Submission introducing new datasets must include the following in the supplementary materials:

   (a) Dataset documentation and intended uses. Recommended documentation frameworks include datasheets for datasets, dataset nutrition labels, data statements for NLP, and accountability frameworks. [We document the code to use the dataset in the repository and provide many references in the Appendices.]

   (b) URL to website/platform where the dataset/benchmark can be viewed and downloaded by the reviewers. [We provide this in Appendix A]

   (c) Author statement that they bear all responsibility in case of violation of rights, etc., and confirmation of the data license. [We provide this in Appendix B and upon submission to OpenReview]

   (d) Hosting, licensing, and maintenance plan. The choice of hosting platform is yours, as long as you ensure access to the data (possibly through a curated interface) and will provide the necessary maintenance. [We provide this in Appendix A]

2. To ensure accessibility, the supplementary materials for datasets must include the following:

   (a) Links to access the dataset and its metadata. This can be hidden upon submission if the dataset is not yet publicly available but must be added in the camera-ready version. In select cases, e.g when the data can only be released at a later date, this can be added afterward. Simulation environments should link to (open source) code repositories. [We provide this in Appendix A]

   (b) The dataset itself should ideally use an open and widely used data format. Provide a detailed explanation on how the dataset can be read. For simulation environments, use existing frameworks or explain how they can be used. [We provide example usage, documentation, and detailed specification in how data is stored in several Appendices]

   (c) Long-term preservation: It must be clear that the dataset will be available for a long time, either by uploading to a data repository or by explaining how the authors themselves will ensure this. [We provide a maintenance plan in Appendix A]

   (d) Explicit license: Authors must choose a license, ideally a CC license for datasets, or an open source license for code (e.g. RL environments). [We provide an open license in Appendix B]

   (e) Add structured metadata to a dataset's meta-data page using Web standards (like schema.org and DCAT): This allows it to be discovered and organized by anyone. If you use an existing data repository, this is often done automatically. [We are in the process of building a site to host, but are happy to do this when completed.]

   (f) Highly recommended: a persistent dereferenceable identifier (e.g. a DOI minted by a data repository or a prefix on identifiers.org) for datasets, or a code repository (e.g. GitHub, GitLab,...) for code. If this is not possible or useful, please explain why. [Code and instructions will always live on GitHub].

3. For benchmarks, the supplementary materials must ensure that all results are easily reproducible. Where possible, use a reproducibility framework such as the ML reproducibility checklist, or otherwise guarantee that all results can be easily reproduced, i.e. all necessary datasets, code, and evaluation procedures must be accessible and documented. [We provide access to code for experiments and will tidy and rehost upon release of the dataset.].

4. For papers introducing best practices in creating or curating datasets and benchmarks, the above supplementary materials are not required.

