# OpenReview forum: "Dungeons and Data: A Large-Scale NetHack Dataset"
_NeurIPS.cc/2022/Track/Datasets_and_Benchmarks — NeurIPS 2022 Datasets and Benchmarks _

### Official Review · Reviewer_qVv5 · 2022-07-11
**Potentially useful dataset**

**Rating:** 7
**Confidence:** 4

**Strengths:**

The dataset can be useful for research in offline RL, but also possibly of interest to the broader RL community. The efficient implementation is definitely a plus, given the computational hunger of current RL methods.

**Weaknesses:**

As the paper indicates, the symbolic method used to gather the data can be characterized as slightly above "Beginner" level. This data is static, in the sense that it is gathered once and is not updated, with larger parts of the state space explored. Thus, at some point, the dataset will become obsolete. Even if that day may be not too close, it will come. Therefore, it would be beneficial to have a sort of symbolic agent that can search the state space and update the dataset with additional trajectories. Obviously, no significant portion of the entire state space can be kept offline, but maybe a symbolic agent would be able to generate some portions on the fly.

**Additional Feedback:**

Questions to the authors:
1. What's the size of the state space of NLE?
2. What is the number of unique states along the trajectories in your database?
3. Is there a high-level symbolic data used by the symbolic agent that can be gathered from the dataset? Factored state representation? High-level actions and their models (e.g., PDDL models)?

**Clarity:**

The paper is very readable. No clarity issues.


**Correctness:**

Not relevant.


**Documentation:**

The dataset is sufficiently documented.


**Ethics:**

I have no ethical concerns related to this work.

**Relation To Prior Work:**

Related work section clearly indicates the difference from prior work.


**Summary And Contributions:**

The paper proposes a dataset of trajectories gathered from the game of NetHack. It has human-generated trajectories (NLD-NAO), as well as ones generated by a symbolic agent that won the NetHack challenge at NeurIPS 2021 (NLD-AA). The paper describes the dataset, its analysis in terms of the game coverage, as well as experimental results for popular RL methods. These include online RL, as well as offline RL and learning from demonstrations using the dataset NLD-AA. The results show the large gap between the current RL methods and the symbolic agent, not to mention the human players.


Minor: You refer to Table 4 multiple times, was that supposed to be Table 2?

---

> ### Author Response · Authors · 2022-08-04
> **Response to Reviewer qVv5**
>
> Dear Reviewer,
> We thank you for the positive review and insightful comments. We were delighted to hear that you qualified our dataset as efficient and useful for the research community. We will be sure to correct our references to Table 4 (which is in the Appendix and is in error) in a future revision. Below we hope to address your concerns and other questions:
>
> > This data is static, in the sense that it is gathered once and is not updated, with larger parts of the state space explored.
>
> It is certainly true that while challenging today, the challenge of  NLD-AA might one day be overcome.  At that point it will be up to researchers to decide whether they need to create a new dataset, if they wish.  One of our goals is to empower future researchers to do this, and to be able to both create and train on their own datasets, should they wish.  They can currently do this with TtyrecDataset object, and examples of how to do this are provided in the supplement info.
>
> > Therefore, it would be beneficial to have a sort of symbolic agent that can search the state space and update the dataset with additional trajectories.
>
> This is a very interesting proposal for research, though perhaps out of scope for this paper. If we understand correctly, you are proposing generating trajectories by using RL agents to play out portion of the episode, and then switching to a symbolic ‘exploratory’ agent to play through and collect the rest of the data for the episode.  While we have not heard of such a method, it sounds like an interesting proposal for further data accumulation that may be useful, when the challenge of NLD-AA  is solved.
>
> > What's the size of the state space of NLE?
>
> This number is almost impossible to quantify since NLE is a highly stateful game. A complete run involves traversing 52 random dungeons, with monsters, items, and topologies that are persistent (once randomly generated) and highly stateful. EG: Monsters have hit points, armor class; items have can be ‘wet’, ‘greased’, ‘broken’, ‘rusted’, ‘poisonous’; the agent itself can has a range of stats, and also can be lucky/unlucky, hungry etc etc.  Much of the state is hidden from the player, making it very difficult to estimate.
> The observation space is 24 x 80 characters, 24 x 80 colors and a cursor, but as a fully fledge role-player game, the state space is vast. We recommend checking out [the NetHack Wiki](https://nethackwiki.com/wiki/Main_Page)
>
> > What is the number of unique states along the trajectories in your database?
>
> As mentioned above the state space of NLE is vast and highly hidden. It is highly unlikely there will be unique states between trajectories (there may be some very similar ones along a trajectory, if one action is a no-op).  If you are requesting about ‘visible states’/observations, we will find out and report these back here.
>
> > Is there a high-level symbolic data used by the symbolic agent that can be gathered from the dataset? Factored state representation? High-level actions and their models (e.g., PDDL models)?
>
> NLD-NAO records only observations, and in NLD-AA only observations, actions and rewards. Any internals state representation used by the symbolic bot is not included in the dataset, as necessarily this could not be stored by the environment.  However, the bot is open source for anyone to use and reproduce, and can be found here: https://github.com/maciej-sypetkowski/autoascend .
>
> Thanks,
>
> Authors

---

### Official Review · Reviewer_EtFb · 2022-07-23
**NLD review**

**Rating:** 7
**Confidence:** 4
**Correctness:** All claims are addressed in this paper.
**Clarity:** This paper is well written.

**Strengths:**

1. The author provides detailed descriptions and analysis of NLD-AA and NLD-NAO, including the performance distribution and other attributes (format, metadata, ...).
2. The author effectively shows the computation cost of this dataset backed with solid numbers (38TB -> 229GB), which is impressive!
3. The author illustrates the difficulty of solving this dataset and posts several promising research directions.

**Weaknesses:**

I did not personally find significant weaknesses.

**Additional Feedback:**

Small typo in appendix:

L540: The branch currently ~currently~ holds ...

A question to author:

Can we use two adjacent states in NLD-NAO to infer the action ((s, s') -> a) by heuristic methods?

**Documentation:**

The documentation overall is great. The repo contains:
- tutorial
- detailed README
- linter
- unit test
- LICENSE

Things could be better if adding more comments in code.

**Relation To Prior Work:**

Yes.

**Summary And Contributions:**

This paper presents NLD (NLD-NAO and NLD-AA), a large-scale dataset of NLE environment. NLD is easy to use (cheap but large-scale, pipeline code ready) and is hard for offline RL community. Several promising directions can utilize NLD, including learning from only observations.

---

> ### Author Response · Authors · 2022-08-04
> **Response to Reviewer EtFb**
>
> Dear Reviewer,
>
> We thank you for the valuable feedback and positive review, which notes that NLD is computationally accessible and can be used to advance several important research directions for RL. We will be sure to correct the typos pointed out in a future revision.
>
> In response to your question:
>
> > Can we use two adjacent states in NLD-NAO to infer the action ((s, s') -> a) by heuristic methods?
>
> This is a very good question and the answer is… “sometimes”.  Some actions result in a well determined top-line message on the screen, or certain menus to open. For instance, the zap action (‘Z’) will either open a menu which says “Choose which spell to cast”, or print “You don’t know any spells right now”.
>
> While one could build a heuristic classifier to infer these, there are multiple challenges: a) first, the problem is under-specified. Certain actions will not present a message, but may change the underlying game state, and it may be impossible to disambiguate these with only two states b) the top line message is highly templated and rich, and so is very challenging to enumerate in its entirety, and probably impossible to enumerate without playing the game. Lastly, c) the exact message is highly context specific. For example, if the character is blind or hallucinating, the message can differ drastically (see https://nethackwiki.com/wiki/Hallucinatory_messages) making it difficult to write a symbolic bot anticipating all the possible special cases.
>
> Given these challenges, a heuristic approach is difficult though, by no means impossible if one is happy to accept highly noisy inference.
>
> Thanks,
>
> Authors

---

> > ### Comment · Reviewer_EtFb · 2022-08-28
> > **Thanks for your feedback**
> >
> > I think my concerns are addressed.

---

### Official Review · Reviewer_Nqea · 2022-07-25
**A Large-Scale Dataset of Demonstrations from NetHack**

**Rating:** 6
**Confidence:** 4
**Correctness:** Yes, the dataset is constructed in a …
**Clarity:** Yes, the paper is well written

**Strengths:**

1.The dataset is combined with NLE, which provides potentially large insights for the study conducted in MDP problem. The proposed dataset enables computationally-accessible research in multiple areas including imitation learning, offline RL, learning from sequences of only observations, as well as combining learning from offline data with learning from online interactions.

2.The dataset has many properties of real-world domains such as partial observability, stochastic dynamics, sparse reward, long trajectories, rich environment, diverse behaviors, and a procedurally generated environment. Such properties allow the dataset to provide a more realistic evaluation environment, thereby making the evaluation of RL algorithms more reliable.

3.NLD can enable agents learn from demonstrations containing only observations, learning from both static data and environment interaction.

4.The proposed NLD is complete, which is decomposed of three components: NLD-NAO, NLD-AA, and TtyrecDataset. The significance of each component for dealing with decision-making problems is clearly explained in the paper. Also, the details of the released data, including the used raw features, file format, and scalability are described clearly.

5.The dataset have been opensourced, and there is a detailed introduction for usage in the corresponding repository, thus facilitating researchers to quickly get started with the development and research of NLD.

6.NLD strikes a better balance between scale (i.e. a large number of diverse human demonstrations on a complex task) and efficiency (i.e. cheap to use and fast to run).

7.Experimental results indicate that NLD poses a substantial challenge to state-of-the-art methods, as the NLE environment is highly stochastic, and partially observed.

**Weaknesses:**

1.The authors should explain the relations between symbolic and RL in details, corresponding to the sentence in Section 2 "Symbolic bots decisively outperformed deep RL methods, with the best performing symbolic bots surpassing state-of-the-art deep RL methods by a factor of 5."

2.The paper should provide a quantitative metric to quantify data scale and efficiency of implementation in reinforcement learning between different datasets. Such metric could further demonstrate the significance of the NetHack dataset. Also, the authors need conduct experiments to compare different decision-making datasets in the area of randomness, magnitude of actions and state spaces, and partial observability. In other words, the paper needs to quantitatively emphasize the necessity and pioneering of NLD for existing research.

3.Section 5 does not fully explain the research significance of this dataset by only mentioning that there is room for improvement in symbolic, i.e., NAO-AA.

**Additional Feedback:**

No.

**Documentation:**

Yes, the dataset is well documented.


**Ethics:**

I have no ethical concerns related to this work.

**Relation To Prior Work:**

Yes, related work section clearly indicates the difference from prior work.


**Summary And Contributions:**

The paper introduces NLD, a large-scale dataset of demonstrations from NetHack. The dataset enables research in multiple areas, such as imitation learning and learning from both offline data and online interactions. Empirical results indicate that significant research advances are needed to leverage large-scale datasets to solve challenging decision-making problems fully.

---

> ### Author Response · Authors · 2022-08-04
> **Response to Reviewer Nqea**
>
> We thank you for your positive comments noting the applicability of the dataset to a wide number of research domains, the challenging real-world properties of NetHack environment, the computational accessibility of the work, and the overall completeness of the project. We also thank you for your thoughtful feedback to improve our paper. We address your comments below:
>
> > 1.The authors should explain the relations between symbolic and RL in details
>
> We apologise for the confusion. In our case a “Symbolic bot” is one whose “behaviours are based on rules that are functions of the symbolic state of the game”, in contrast to RL methods where “behaviours are learnt through iterative interaction with the game (training)”. Naturally, the definition is fluid here - it is possible to have neurosymbolic or hybrid models too.
>
> The winning Symbolic bot of the NetHack Challenge won by coding up sophisticated sets of behaviours into strategy-like subroutines and chose between these strategies using a priority ranking. This approach beat previous Deep RL methods by a factor of 5 but is still extremely far from winning at NetHack. More details are available in the Challenge report (Hambro et al 2022)
>
> >  2.The paper should provide a quantitative metric to quantify data scale and efficiency of implementation in reinforcement learning between different datasets.
>
> Note that Appendix J – Table 7 provides a quantitative comparison of the scale and efficiency of our dataset relative to StarCraft and MineRL (the two most related and large-scale datasets we are aware of). Here we define data scale as ‘number of trajectories’, and efficiency as throughput of frames per second (FPS) for both the dataset and environment. Our dataset of human trajectories, NLD-NAO is significantly larger than MineRL and about as large as StarCraft, while also being significantly faster to run. In addition, the NetHack environment is significantly faster than its peers in StarCraft and MineRL. The reason for this is that the StarCraft and MineRL datasets both require the core game engines to ‘play through’ and render their saved game replays - a measure that trades off speed against memory (replays can be smaller, but are slower to render).  By virtue of its symbolic, terminal-based interface, NetHack requires no such trade off -  replays can be both small, and fast to load.
>
> > Also, the authors need conduct experiments to compare different decision-making datasets in the area of randomness, magnitude of actions and state spaces, and partial observability.
>
> While we agree that such complexity measure would be useful, we are not aware of any such metrics in the literature. If the reviewer can point us to such metrics, we are happy to incorporate them in our analysis.
>
> NetHack is a POMDP, where the observation is a tiny fragment of the overall world - merely what's visible in the current dungeon. The entire world, with all the item state, maps, monsters, player hidden-state etc. persists once generated, resulting in a rich world where a decision taken in the first few steps, can have a significant impact thousands of steps down the line.
>
> Furthermore, NetHack’s procedural generation is sophisticated - the entire world is dynamically generated based on the _current state_ of a random number generator seed.  This state will advance any time sampling occurs (which is often).  In contrast, for example, in OpenAI’s ProcGen environment all level generation is based on the _initial state_ of a random number generator seed, meaning that levels are effectively determined by the initial seed. The end result is that NetHack has a much larger ‘branching factor’ in terms of the worlds it generates. NetHack has great randomness… but quantifying it is hard!
>
> For more see (Kuttler et al. 2020).
>
> > 3.Section 5 does not fully explain the research significance of this dataset by only mentioning that there is room for improvement in symbolic, i.e., NAO-AA.
>
> We believe this dataset highlights a number of important challenges for research.
>
> First, we demonstrate that methods for offline RL, imitation learning, and learning from sequences of observations struggle to learn well from our datasets. This indicates a need to develop more effective methods for highly stochastic, partially observed environments with vast state spaces. NLD also opens up this research for learning in the ‘internet scale’ regime, where sufficient data is available to train foundation models, to investigate the formulation of curricula, or to investigate the self-supervised learning of trajectories. Finally, NLD democratizes research in these areas and allows researchers to generate and train on large-scale datasets with limited resources, through it its efficiency/compression.
>
> We hope the clarifications above are sufficient for you to reconsider your assessment of the paper. Please let us know if you have any outstanding concerns that stand between us and a stronger recommendation for acceptance.

---

> > ### Author Response · Authors · 2022-08-04
> > **References!**
> >
> >
> > [1] Insights from the NeurIPS 2021 NetHack Challenge (Hambro et al 2022)
> >
> > [2] The NetHack Learning Environment (Kuettler et al 2020)

---

> > ### Comment · Reviewer_Nqea · 2022-08-28
> > **Thanks for your feedback**
> >
> > I think my concerns are addressed.

---

### Official Review · Reviewer_YgJb · 2022-07-26
**A large-scale dataset on a niche problem**

**Rating:** 7
**Confidence:** 4

**Strengths:**

* The paper is well-written and easy to follow.
* The human dataset is high quality, as it is scraped from actual game plays on the web servers. The scale is huge - 1.5M human trajectories with almost 10B transitions. This is much more than most other video game benchmark data, and is provided under the open-source GPL license.
* The NLD-AA dataset contains trajectories generated by a winning bot on NetHack. While not as good quality as human dataset, it has groundtruth actions to train inverse dynamics model and do direct behavior cloning. It is complementary to NLD-NAO and I am glad that the authors included both.
* The experiments use off-the-shelf methods like A-PPO, DQN, CQL, etc. No novel algorithm is proposed, but these methods are standard enough to provide good baseline results for the dataset.
* "Dungeon and Data" - great name!

**Weaknesses:**

My main concern is about the benchmark's contribution to advances in the broader policy learning community. NetHack is a very niche domain. It is an ASCII-based game that does not make much sense to untrained eyes, and does not have any meaningful high-dimensional observation like 3D perception or other sensing modalities. In addition, the action space and world transition dynamics are quite simplistic. I am not convinced that the potential future algorithms developed on NetHack or NLD will be generally applicable to other embodied agent domains. As the authors pointed out, even a purely hard-coded agent called "AutoAscend" is able to achieve nontrivial performance on the benchmark. AutoAscend bot actually contributes the NLD-AA dataset with full state-action-score trajectories.

I am not claiming that mastering NetHack is easy. But in contrast, it is extremely difficult to hard-code a robot agent in Habitat [1] or AI2Thor [2] from pixels alone even for the simplest tasks. While the authors discussed connection to robotics in L337-339, I am still doubtful of NetHack as an effective testbed for general-purpose embodied agent algorithms.

That being said, I'm still leaning towards acceptance for the large-scale dataset introduced in this paper, considering that much simpler Gridworld environments without datasets have been accepted at top conferences.

* [1] Habitat: A Platform for Embodied AI Research. Savva et al. 2019.
* [2] AI2-THOR: An Interactive 3D Environment for Visual AI. Kolve et al. 2019.

**Additional Feedback:**

Fig. 1 is a bit hard to read and could be improved. Maybe something like a violin plot can be better: https://en.wikipedia.org/wiki/Violin_plot

**Clarity:**

The paper is clear to read and easy to follow. The details of the dataset and experiments are thoroughly explained.

**Correctness:**

As far as I am concerned, it is constructed correctly and contains enough details to reproduce. The experiments use standard methods like DQN, A-PPO, CQL, and BCO, and have sound details.

**Documentation:**

Yes, the supplementary material includes hosting & maintenance plan, licensing, details on dataset format, metadata, and data collection details. There are no ethical and responsible use concerns.

**Ethics:**

No.

**Relation To Prior Work:**

The paper discusses prior works in depth on offline RL benchmarks and large-scale human demonstration datasets. In my opinion, it has discussed the differences in a satisfactory manner.

**Summary And Contributions:**

This paper introduces the Nethack Learning Dataset (NLD), which is a large dataset of demonstrations from the NetHack game. It has 2 partitions: NLD-NAO that contains 10 billion state-only trajectories and metadata from 1.5M human games, scraped from online web servers; and NLD-AA, a collection of 3 billion trajectories with complete state, acton, and scores, generated by a winning bot on the NetHack challenge. Python scripts are provided to load these datasets efficiently. Experimental results are demonstrated on the dataset with RL methods like APPO and DQN, and imitation learning methods like BC and BCO (clone from observations).

---

> ### Author Response · Authors · 2022-08-04
> **Response to Reviewer Ygjb**
>
> Thank you for your positive review and valuable feedback. We are glad to hear you found the paper well-written and easy to follow, and the dataset of high quality with sufficient baselines.
>
> We respectfully but decisively disagree with the NetHack Learning Environment or the NetHack Dataset being niche. Yes, the game (NetHack) is niche, but as argued in Kuettler et al (2020)[1] and since then experienced in multiple follow on works (including Samvelyan et al (2021) [2],  Hambro et al (2022) [3], Izumiya et al (2020) [4] and many more since) it exemplifies open research questions that generally capable agents need to address. In the context of the NetHack Dataset, we believe despite its simple visual appearance, it will allow researchers to specifically focus on how to do third-person imitation learning for a vast and complex (yet visually simple) environment where a significant discrepancy between expert trajectories and an agent’s own experience has to be expected due to the procedural content generation of NetHack. We argue that this is a phenomena present in most real-world problems for RL and we don’t see a reason why methods attempting to address this research question on the NetHack Dataset should not provide us with insights about how to generally deal with such problems in the future. The fact that NetHack and this dataset are visually simple is a strength, not a weakness, as it allows research to separate concerns and tackle research problems (e.g. third-person imitation learning, inverse RL) without overemphasizing an agent’s visual understanding. Furthermore, the sheer mass of human demonstrations and the diversity of human behaviors in this dataset is unprecedented. In addition, compared to Habitat or AI2-THOR, the action space is vast, which will lead to very different research focus of future work on this benchmark.
>
> [1] The NetHack Learning Environment (Kuettler et al 2020)
>
> [2] MiniHack the Planet: A Sandbox for Open-Ended Reinforcement Learning Research (Samvelyan et al 2021)
>
> [3] Insights from the NeurIPS 2021 NetHack Challenge (Hambro et al 2022)
>
> [4] Attention-based inventory management (Izumiya et al 2021)

---

> > ### Comment · Reviewer_YgJb · 2022-08-19
> > **Raising my score!**
> >
> > Thanks for the detailed and eloquent response. I believe this dataset will be valuable to the policy learning community, and firmly advocate for this paper's acceptance. I have raised my score accordingly.

---

### Official Review · Reviewer_PXHE · 2022-07-26
**Human and a champion bot play trajectories for Nethack Learning Environment.**

**Rating:** 8
**Confidence:** 4

**Strengths:**

NLD-NAO collects 1.5M human game plays in a state (only) trajectories form with game play metadata. NLD-AA collects 100K plays by the winning bot (AutoAscend) from a NeurIPS-21 competition, in the state-action-score trajectory form.  NLE itself seems to be updated (v0.9.0) to be compatible with NLD, containing TtyrecDataset in its python modules (nle.dataset).

**Weaknesses:**

It is essentially valuable that the work includes experiments for online RL, offline RL, imitation learning, and learning from demonstrations, which demonstrate the example usage and usefulness of the dataset.  Unfortunately, the process for reproducing experimental results shown in the paper is not clear from the main paper, supplemental material, or github (https://github.com/dungeonsdatasubmission/dungeonsdata-neurips2022/tree/a3f01c425f5d75ce9174b76105ac34a377c2df30/experiment_code).

**Additional Feedback:**

Minor errors here and there.

For example,
Appendix p.2 564   before the an => before an
565 are may => may

Please go through proof-reading services.

**Clarity:**

It is well written to cover the details well in general.  Proof-reading is must, though.

**Correctness:**

TtyrecDataset is released as a part of the updated NLE (v0.9.0), not the dataset repository.  It is totally ok as the work but please make it clear and claim what is NLE update and how it is available as well as the dataset release.

**Documentation:**

Description about dataset itself is great.  Experimental reproducibility should be addressed more so that anyone can reach to the experimental results shown in the paper to exercise with the dataset.

**Ethics:**

Not applicable.

**Relation To Prior Work:**

Comparing to related dataset for relevant benchmarks (or gym) of StarCraft, Dota, and MineRL is good and sufficient to this level, the reviewer believes.  Starcraft ii: A new challenge for reinforcement learning. ArXiv, abs/1708.04782, 2017.  Dota 2 with large scale deep reinforcement learning. ArXiv, abs/1912.06680, 2019. Minerl: A large-scale dataset of minecraft demonstrations. ArXiv, abs/1907.13440, 2019.

**Summary And Contributions:**

The paper introduces a dataset called NLD, to be useful for imitation learning and offline RL study for Nethack Learning Environment (NLE) [Kuttler+, NeurIPS-20]. NLE (and potentially for a controlled version of NLE, MiniHack [Samvelyan+, NeurIPS-21]) is an OpenAI gym environment from the same author group, with a popular Rogue-like (or dungeon explorer-type) game, NetHack.

NLD-NAO collects 1.5M human game plays in a state (only) trajectories form with game play metadata. NLD-AA collects 100K plays by the winning bot (AutoAscend) from a NeurIPS-21 competition, in the state-action-score trajectory form.  NLE itself is updated (v0.9.0) to be compatible with NLD.

Experiments using the dataset is included.  Related dataset for similar benchmarks (StarCraft, Dota, and MineRL) is discussed.

---

> ### Author Response · Authors · 2022-08-04
> **Response to Reviewer  PXHE**
>
> Dear Reviewer,
>
> We thank you for the positive review and helpful feedback. We are delighted that you think our work is well written, clear and valuable in comparison to existing work in the literature.
>
> > Unfortunately, the process for reproducing experimental results shown in the paper is not clear from the main paper, supplemental material, or github (https://github.com/dungeonsdatasubmission/dungeonsdata-neurips2022/tree/a3f01c425f5d75ce9174b76105ac34a377c2df30/experiment_code).
>
> Thank you for pointing this out! While the README.md in the directory linked does have install instructions, and configs necessary for the experiments, we take on board the point and will add a script to specifically reproduce all the experiments used in the paper.
>
> > TtyrecDataset is released as a part of the updated NLE (v0.9.0), not the dataset repository. It is totally ok as the work but please make it clear and claim what is NLE update and how it is available as well as the dataset release.
>
> Thank you for the suggestion! We are currently in the process of setting up the permanent website at ai.facebook.com/datasets. We will make it clear on that site, and on NLE, how the two relate to each other.
>
> > Minor errors here and there
>
> Thank you for spotting these. We will certainly proof read again and fix these up in a future revision.
>
> Thank you once again for your consideration,
>
> Authors

---

### Official Review · Reviewer_kge3 · 2022-07-26

**Rating:** 8
**Confidence:** 3

**Strengths:**

1. The paper presents a large-scale dataset for a fast and challenging RL environment (NetHack). The authors also provide high-performance code to load this dataset and train agents, which makes larger-scale experiments more accessible to those with smaller compute budgets.


2. The dataset and code is well-documented.

3. Baseline results cover a range of different approaches to train agents (online RL, offline RL, learning from observations only, combinations of offline and online learning).


**Weaknesses:**

1. Appendix G.6 mentions a SQL interface to consider subsets of the datasets, according to the metadata. Based on Table 2, the performance of learned policies is quite below the dataset average.
I think the paper would benefit from additional investigation and discussion on why agents trained on NLD-AA (more narrow data distribution generated by symbolic bot, cleaner demonstrations) outperform those trained on NLD-NAO (human demonstrations).
For instance, cutting NLD-NAO trajectories short so the length of the trajectories are similar to NLD-NAO to exclude the later-stage states of NetHack. From Figure 1, it seems that NLD-NAO trajectories may be several times longer than NLD-AA.

2. Only 5 seeds are reported. In Figure 2, it seems that APPO methods have very high variance (blue and orange lines), and would likely benefit from running more seeds. The discussion would also benefit from exploring why APPO has higher variance b/w 0 and 600M steps, after which the variance suddenly drops.


**Additional Feedback:**

Minor comments:

L52-57: There is too much text formatting (bold / italics) that makes the paragraph difficult to read.

Figure 1 caption: Centered text alignment looks strange.

Table 2: What is the difference b/w “-” and “(see col 1)”? Please clarify in the caption.

Figure 2: What does “- @” mean?

The paper uses both “BCO” and “BC(O)” to refer to Behavioral Cloning from Observations?


**Clarity:**

Yes, the paper is well-written and clear. I have a few minor comments in Additional Feedback which can be easily corrected.


**Correctness:**

To my knowledge, yes.


**Documentation:**

Yes.


**Ethics:**

No further review needed, the authors have sufficiently covered any potential concerns in the Appendix.


**Relation To Prior Work:**

Yes, I think discussion of prior work is adequate. The paper compares NLD to related large-scale datasets for Starcraft, Dota, and MineRL, which require significantly more compute to use.


**Summary And Contributions:**

This paper presents the NetHack Learning Dataset (NLD), which has 3 parts: i. 1.5 million human trajectories recorded from the NAO public NetHack server; ii. 100,000 trajectories from the symbolic bot winner of the NetHack challenge 2021; iii. code for users to apply these trajectories in a compressed format. To demonstrate the utility of NLD, the authors train and compare several algorithms spanning online RL, offline RL, imitation learning, and learning from observations only.

---

> ### Author Response · Authors · 2022-08-04
> **Response to Reviewer kge3**
>
> Dear Reviewer,
>
> We thank you for your thoughtful comments, positive feedback, and the time taken to review which have helped to further strengthen our work. We hope to address your remaining concerns below.
>
>  > Appendix G.6 mentions a SQL interface to consider subsets of the datasets, according to the metadata. Based on Table 2, the performance of learned policies is quite below the dataset average. I think the paper would benefit from additional investigation and discussion on why agents trained on NLD-AA (more narrow data distribution generated by symbolic bot, cleaner demonstrations) outperform those trained on NLD-NAO (human demonstrations).
>
> As mentioned in Section 6.2, NLD-NAO contains only states and no actions. In order to use this dataset, we infer the actions (given two consecutive frames) using trajectories generated by a random policy. Given the vastness of the state space and the hard exploration challenge posed by NetHack, we expect this policy to only visit a very small fraction of the entire state space. Thus, our inferred actions won’t be very accurate, particularly for later parts of the game. Note that this problem is also highly underspecified since in many cases some actions have little effect on consecutive frames, which makes it even more difficult to learn an accurate inverse model.
>
> In contrast, NLD-AA contains state-action pairs which makes it much easier to approximate the behavior of the symbolic bot. However, as you note, the performance of our methods is still below the corresponding dataset average, which further emphasizes the challenge posed by this benchmark and its potential to drive future research on more effective methods for learning from demonstrations. We believe this is largely due to NetHack’s high level of stochasticity, partial observability, and large state space. We will update the paper with more discussion about the different challenges of learning from these two datasets. Overall, we believe NLD-NAO exemplifies the difficulty of making offline datasets without actions useful for RL. The fact that current learned policies are below NLD-NAO average emphasizes the need for concentrated future research on how to utilize such demonstrations—something we hope our benchmark will facilitate.
>
> > For instance, cutting NLD-NAO trajectories short so the length of the trajectories are similar to NLD-NAO to exclude the later-stage states of NetHack. From Figure 1, it seems that NLD-NAO trajectories may be several times longer than NLD-AA.
>
> Our paper focuses on introducing the two datasets and evaluating a range of widely-used baselines on it, so our aim wasn’t to introduce a novel / more effective approach. Nevertheless, we agree with the reviewer that this is a promising direction for future research. Indeed, we believe there are a number of interesting ways one could adapt the dataset to improve the performance of a learner such as truncating episodes, creating or adapting a curriculum, or simply improving the action inference (perhaps using a larger model over the whole sequence). While these are avenues we would love to see pursued, we feel this is best left to the ingenuity of the community instead of being marked as a core contribution. We would love to see this work!
>
> > Only 5 seeds are reported. In Figure 2, it seems that APPO methods have very high variance (blue and orange lines), and would likely benefit from running more seeds. The discussion would also benefit from exploring why APPO has higher variance b/w 0 and 600M steps, after which the variance suddenly drops.
>
> We have indeed noted the early variance between 0 and 600M steps, which drops later in training. This phenomenon (which we have noted with NetHack and refer to internally as ‘Taking Off’) corresponds to when agents learn behaviours that unlock more of the game and allow them to rapidly increase their short term performance. Since the game is procedurally generated and has a vast state-action space it is hard to predict when such experiences will occur. Examples of such behaviour that can drastically impact the length of the episode are ‘learning to kick down locked doors’, ‘learning to pray’, ‘learning not to kill your pet’. We believe such a ‘Taking off’ phemonenon occurs in many kinds of environments.  In the interest of better illustrating this phenomenon, and reducing variance, we plan to run 10 seeds and run these for longer, adding these to the final camera ready, and plotting individual runs.
>
> > Minor comments: [...]
>
> Thank you for spotting these, we will update and clarify these in a future revision.
> We hope the clarifications above are sufficient for you to strongly support our paper. Please let us know if you have any outstanding concerns that stand between us and a strong recommendation for acceptance.
>
> Thank you once again for your consideration,
>
> Authors

---

### Author Response · Authors · 2022-08-29
**Thank you for Reviews**

Thank you to all reviewers for your comments and positive feedback.  We have updated the revision slightly with a plot of sample runs (instead of error bars), a comment on the ``taking off'' phenomenon as addressed in the reviews & comments, and minor fixes as requested.  As mentioned in our review comments, we will update the camera ready with data from 10 seeds.  We thank you all for your efforts in helping us further improve the paper, and your time taken to review.

---

### Meta-Review · Area_Chair_2QqG · 2022-09-12

**Recommendation:** Accept
**Confidence:** 4

**Metareview:**


This paper presents the NetHack Learning Dataset (NLD), which has 3 parts: i. 1.5 million human trajectories recorded from the NAO public NetHack server; ii. 100,000 trajectories from the symbolic bot winner of the NetHack challenge 2021; iii. code for users to apply these trajectories in a compressed format. To demonstrate the utility of NLD, the authors train and compare several algorithms spanning online RL, offline RL, imitation learning, and learning from observations only.

weakness:
- A niche domain: "NetHack is a very niche domain. It is an ASCII-based game that does not make much sense to untrained eyes, and does not have any meaningful high-dimensional observation like 3D perception or other sensing modalities"
- quantitative metric to quantify data scale and efficiency of implementation
- This data is static, in the sense that it is gathered once and is not updated, with larger parts of the state space explored.

Some of these points were addressed in the rebuttal, while the challenge of a static dataset is deferred to future work.
Broadly there is agreement among the reviewers, and the ACs that this is a useful benchmark for the community.

AC would request the authors to carefully integrate all the feedback in the updated manuscript as well as any leftover comments added as clarifications in the appendix.

---

### Decision · Program_Chairs · 2022-09-16

Accept